# Effects of Lake Productivity on Density and Size Structure of Pelagic Fish Estimated by Means of Echosounding in 17 Lakes in Southeast Norway

**DOI:** 10.3390/s21103391

**Published:** 2021-05-13

**Authors:** Arne N. Linløkken

**Affiliations:** Faculty of Applied Ecology, Agricultural Sciences and Biotechnology, Inland Norway University of Applied Sciences, N-2418 Elverum, Norway; arne.linlokken@inn.no

**Keywords:** fish species, species characteristics, zooplankton, predation, algae, phosphorus, development

## Abstract

Density estimation of pelagic fish was performed by means of single beam echosounding in 17 lakes within a period of 34 years, from 1985 to 2018. Surveys were performed repeatedly (two to fourteen times) in five lakes. The density estimates ranged from 34 to 4720 fish/ha and were significantly correlated with total phosphorus concentration. The high density in relatively phosphorus rich lakes (*TP* > 10 µg/L) was comprised of small fish (<20 cm) and was partly due to the higher number of pelagic fish species. The number of pelagic species varied from one, Arctic charr, in the most elevated and oligotrophic lakes, and whitefish dominated in less elevated oligotrophic lakes. In lowland lakes characterized as mesotrophic or tending to mesotrophy, smelt, vendace, and two to three cyprinids comprised the pelagic fish stock. These fish species predate zooplankton effectively, and species composition and body size of planktonic cladocerans was affected by fish density. Large species of *Daphnia* were lacking in lakes with high fish density, and body size of present species, *D. galeata, D. cristata*, and *Bosmina* spp. were negatively correlated with pelagic fish density.

## 1. Introduction

The pelagic fish community plays an important role in lake ecology and also serves as a human food resource [1,2]. Exploitation of freshwater fish stocks has been increasing in developing countries but has been declining in industrialized societies during the last decades [3,4,5]. Reduced fishery may lead to increased fish density, and furthermore to increased competition for food and reduced individual growth and size of fish [6,7,8]. In parallel to this, temperature increase and shorter winters may affect fish density as the recruitment of lake-dwelling fish species is often positively correlated with temperature [8,9,10,11,12]. The typical plankton-predating salmonids vendace (*Coregonus albula*), whitefish (*Coregonus lavaretus*), and smelt (*Osmerus eperlanus*) and the cyprinids roach (*Rutilus rutilus*) and common bleak (*Alburnus alburnus*) affect the zooplankton community by grazing the herbivorous species, especially the large species of the crustacean genus *Daphnia* [13,14,15,16], when available. Herbivorous plankton species may control algae growth [17,18,19,20], which in nutrient-rich lakes may excide the biomass amount that can be further traded either through zooplankton or by aerobic bacterial degradation in the water masses [21,22,23]. Oxygen deficit and consequently poor conditions for aerobic organisms in the lake’s deep layers may follow, with negative effects on the water quality regarding most kinds of public use [24,25]. Increased density of planktivorous fish combined with the fish being smaller in size, may therefore affect the lake’s water quality negatively. This may be enforced by the dominance of smaller sized fish affecting the abundance of the large algae-feeding zooplankton species by predating their young stages [18,26,27], and both fish and zooplankton should be monitored routinely.

The pelagic fish density can be effectively estimated and monitored by means of echosounding [8,28,29,30]. The method gives an estimate of the number of pelagic fish per ha and size distribution of the fish stock based on the echo strength along selected transects. By recording along a reasonable number of transects in a lake, preferably over areas with depths greater than 10 m, an average density of fish in the pelagic habitat can be calculated. If the density of transects or between sections of the lake varies largely, the accuracy will be low (i.e., high variance) and more transects should be sampled. The shape of the lake, the fish species composition, the time of year and the day affect the proportion of the fish stock being pelagic, and echosounding will give a minimum estimate of the total fish abundance in the lake. There is considerable uncertainty associated with the method and comparing the fish density of lakes that are relatively similar (i.e., oligotrophic lakes with the same fish species), may be difficult. The difference between oligotrophic and meso- or eutrophic lakes and between lakes with different species composition, on the other hand, is normally so pronounced that it can be easily revealed by echosounding.

This study included 17 lakes in southeastern Norway, from the Halden River system in southeasternmost Norway with relatively productive lakes, to the upper parts of the Glomma and the Trysil/Klarälv River systems with oligotrophic lakes in the north, and three lakes in the Drammenselva River system west of the Glomma River system. There are substantial differences in fish community composition, partly due fish to immigration history [31], differing water quality, and climatic conditions, and 10 of the lakes are regulated for hydropower production. Pelagic fish density and size were estimated and related to lake productivity and zooplankton community. The following hypothesis were tested:

Fish density increases with lake productivity (Tot-phosphorous, *TP*).

The structure of the zooplankton community is affected by fish density.

Echosounding was performed to estimate density and size structure of pelagic fish, and species present were in part based on previous and in part on simultaneous sampling with pelagic gill nets. In three lakes, the echosounding was performed repeatedly at different times of the year and time of day to assess the effect of timing on the abundance of pelagic fish (i.e., on the spatial distribution of fish). Description of water quality, planktonic algae, and zooplankton communities, mainly based on literature studies, were used to describe the lakes, and were related to fish density. 

## 2. Materials and Methods

### 2.1. Study Area

The 17 studied lakes (outlets 61°52.8′ N, 9°45.9′ E–59°19.6′ N, 12°4.9′ E) drain to five different river systems. Lakes Øymarksjøen, Rødenessjøen, and Hemnessjøen drain to the Haldensvassdraget River system in Viken County (Figure 1), Lake Rømsjøen, also in Viken County, drains to the Oselva River and eastward to Sweden. Lakes Randsfjorden, Sperillen, and Dokkfløy drain to the Drammenselva River system, west of Lake Mjøsa, and the Glomma River system. Lakes Einavatn, Gopollen, Furusjøen, and Rondvatnet drain to Lake Mjøsa and further to the Glomma River, and Lakes Storsjøen in Odalen, Osensjøen, Storsjøen in Rendalen, and Atnsjøen are all situated in Innland County and drain to the Glomma River system. Lakes Engern and Sølensjøen, also in Innland County, drain to the Trysilelva River and further to the Klarälven River system in Sweden. 

The lakes are situated 107 to 1167 m a.s.l., and the surface areas range from 0.96 to 140.7 km^2^ (Table 1). The conductivity, total phosphorus (*TP*), and total nitrogen (*TN*) ranged from 0.42 to 9 mS/m, from 3 to 23 µg/L and from 117 to 1200 µg/L, respectively. Nine lakes are moderately regulated, from 0.9 to 6.6 m, whereas Lake Dokkfløy is regulated 65 m. Pelagic fish species vary substantially between lakes as Arctic charr (*Salvelinus alpinus*) is the pelagic species in mountain lakes, whitefish (*Coregonus lavaretus*) occur in lakes at lower altitudes, in some lakes in sympatry with Arctic charr, vendace (*Coregonus albula*), and/or smelt (*Osmerus eperlanus*), and an increasing proportion of cyprinids, with roach (*Ruttilus rutilus*) as the most widespread downstream in the river systems, and common bleak (*Alburnus alburnus*), bream (*Abramis brama*), and white bream (*Blicca bjoerkna*) occur in lakes in the southeastern lowland. Smelt are commonly smaller than 15 cm, cyprinids, except bream, are mostly smaller than 20 cm, vendace are smaller than 25 cm, and Arctic charr are mostly smaller than 30 cm, whereas whitefish may exceed 40 cm. Predatory fish species are brown trout (*Salmo trutta*) in the Arctic charr and coregonid lakes, and pike (*Esox lucius*) and perch (*Perca fluviatilis*) are present in all lakes harboring coregonids and cyprinids, except in Lake Dokkfløy and Lake Gopollen (whitefish are artificially stocked). In Lake Sølensjøen, the population size of whitefish was estimated by means of mark-recapture experiments in 1986 and 1993, showing 41 [54] and 85 [55] whitefish larger than 30 cm per ha, respectively (increased due to reduced exploitation). These figures serve as a basis for comparison.

Most of the lakes are oligotrophic, although Lake Hemnessjøen is mesotrophic and Lake Øymarksjøen, Lake Rødenessjøen, and Lake Storsjøen in Odalen may be border cases, at least in exceptional years. The available data on algae biomass showed 38–518 mg Ww/m^3^ (Table 2) in the ologotrophic lakes, 734 in Lake Storsjøen in Odalen and 1818 mg Ww/m^3^ in Lake Hemnessjøen. According to Brettum [56], the latter may be characterized as mesotrophic (>1200 mgWw/m^3^) whereas Lake Storsjøen in Odalen borders mesotrophic (700–1200 mgWw/m^3^). The zooplankton biomass comprised 0.06–2.8 gDw/m^2^, and the presence of the *Daphnia* species varied with the fish species of the lakes (Table 3). Normally, algae biomass has a peak in early summer (June/July), declining later in summer due to nutrition limitation and eventually predation from zooplankton, which peaks later in August/September. The maximum values of algae and zooplankton may express the potential of the lake.

### 2.2. Sampling and Analysis

The pelagic fish density was estimated irregularly over 34 years from 1985 with a single beam echosounder SIMRAD EY-M [57] in the period 1985–2014, and from 2015 to 2018, the single beam SIMRAD EK15 device [58] was used. The two systems were compared in a survey in Lake Storsjøen in Rendalen in 2016, and the results proved comparable [30].

Sampling was done along transects crossing from shore to shore (see Linløkken et al. [8,30]), and the degree of coverage S (=Pooled recorded distance/(Lake area)^0.5^) was larger than 3, as recommended by Aglen [61], except in three lakes, Lake Hemnessjøen (S = 2.55), Lake Gopollen (S = 2.0), and Lake Furusjøen (S = 0.6), where less than 30% of the surface covered depth >10 m. Fish biomass was calculated by means of the relationship between fish length (*L*) and weight (*W*), expressed as *W* = a • *L*
^b^ [62] with species specific parameters a and b. Depending on the species present in the lake, parameters of smelt (a = 1.7 × 10^−6^, b = 3.19, from Storsjøen in Rendalen, *unpublished*) were used for fish smaller than 15 cm, parameters of vendace (a = 24.0 × 10^−6^, b = 2.26, from Lake Osensjøen [8]) were used for fish smaller than 25 cm, and otherwise parameters of whitefish (a = 15.2 × 10^−6^, b = 2.89, from Lake Osensjøen [8]) or of Arctic charr (a = 6.25 × 10^−6^, b = 3.05, from Lake Sølensjøen [54]) were used.

The EY-M and EK15 echo sounders transmit sound at different frequencies, 70 and 200 kHz, respectively, and the transducer of EY-M had a beam angle of 11.2°, whereas EK15 had a beam angle of 9°. The transducer was directed vertically from 1–1.5 m below the surface toward the bottom, except in Lake Furusjøen, where the transducer was directed horizontally due to the large proportion of shallow areas. The equipment was calibrated from the boat with a calibration sphere at an 8 m depth. For the EY-M calibration, a 32.1 mm copper sphere (corresponding to echo target strength TS = −39.4 dB at temperature 5 °C) was used, and a 38.1 mm wolfram carbide (WC) sphere (corresponding to TS = −39.2 dB at temperature 5 °C) was used for the EK15. Echo or target strength (TS, dB) was transformed to fish length according to this regression: *TS* (dB) = 20 • log (*L*, cm) − 68 [57].

SIMRAD EY-M operates with a fixed pulse duration (PD) of 0.60 ms, and a ping repetition frequency (PRF) of 1.5 or 3.0 pings/s, and 3.0 pings/s was used except for the deepest lake, Storsjøen in Rendalen. This was due to problems with the signal meeting the second bottom echo (the bottom echo of each signal was detected twice, or even more, because the echo was reflected from the surface) of the former signal due to a long delay in the deep lake. The SIMRAD EK15 has several options for PD and PRF, and PD was set to 0.32 ms as this was assumed to detect single fish most accurately in the fish densities of the studied lakes. PRF was set to 4.0 pings/s, except for in Lake Storsjøen in Rendalen, where PRF was set to 2.0 pings/s to avoid the problem above-mentioned.

The water chemistry and plankton community descriptions were mostly based on the open access base Vannmiljø, which is managed by the Norwegian Environment Agency, and on technical reports from the rich archive of the Norwegian Institute of Water Research (NIVA). In particular, data on plankton were based on NIVA. Indices of quantitative algae and qualitative zooplankton characterization were based on Brettum [56] and Løvik [63], respectively. Some supplementary samples of zooplankton were taken during the occasions of echosounding in 2015–2018 (referred to as unpublished), and some were selected from different available reports, which are referred to.

### 2.3. Data Treatment and Statistical Analysis

SIMRAD EY-M stores the data on magnetic tape, and the tapes were later digitalized and analyzed by means of the HADAS software [64]. SIMRAD EK15 was run by a pc with SIMRAD ER15 software to control and sort the data. The EK15/ER15 system stored the raw data, and these were later analyzed with the Sonar5-Pro software [65]. The Sawada index Nv [66] warning limit was set to Nv < 0.1 (default, Sonar5-Pro [65]), and no warnings were received. For EY-M, echoes with duration <2 relative to PD and ≤12 hits (fixed) on the target were considered as single fish [64]. In EK15, single fish detection was set up in this way; echo length 0.7–1.3 relative to PD, medium strength of multiple peak criteria (no dip greater than 1.5 dB within the echo), and maximum gain compensation of 3 dB one way, and a 40 LogR threshold model was applied. Gain was set to 8 on the EY-M, and time variable gain (TVG) was set to 40 LogR (single fish detection) in EK15. The measured distribution of the peak voltage response from single fish echo data were deconvolved by means of a modified Craig and Forbes algorithm to remove the beam pattern effect due to the single beam character.

The density was analyzed in one segment including the depth from 2 m down to 15 to 50 m, depending on the fish distribution at the time of recording. Fish density of the pelagic zone of lakes were calculated as means of single transects, and 95% confidence limits were calculated on log transformed data y = ln(x), assuming negative binomial distribution, and 95% confidence interval of transformed values:

C.I._y_ = ȳ ± t_0.025_ • S.E. For untransformed x: Lower C.L._x_ = mean x • (e^(Lower C.L.y)^/e^ȳ^) and Upper C.L._x_ = mean x • (e^(Upper C.L.y)^/e^ȳ^), according to Elliot [67]. In samples of 0 observations, y = ln (x + 1) transformation was used.

The r software [68] was used to run the following linear models to explore the relationships between pelagic fish abundance (N/ha, B/ha), size distribution expressed as median length (*L*_M_), and environmental factors (*TP, Conductivity,* and *altitude*):*Pelagic fish density* (N/ha) = a + b_1_ • *Conductivity* (µ S/cm) + b_2_ • *Altitude* (m a.s.l.) + *e*
*Pelagic fish biomass* (kg/ha) = a + b_1_ • *Conductivity* (µ S/cm) + b_2_ • *Altitude* (m a.s.l.) + *e*
*Pelagic fish density* = a + b_1_ • *TP* (µg/L) + b_2_ • *Altitude* (m a.s.l.) + *e*
*Pelagic fish biomass* = a + b_1_ • *TP* (µg/L) + b_2_ • *Altitude* (m a.s.l.) + *e*
*Pelagic fish density* = a + b_1_ • *TP* (µg/L) + b_2_ *TP*: *L*_M_ (cm) + b_2_ • *Altitude* (m a.s.l.) + *e*
where a and b_1–2_ are parameters under estimation and *e* is the error assumed to be normally distributed (central tendency as the variable *density* is based on mean values of the transect densities). To explore the potential effects of fish density (i.e., predation) on important planktonic food items, the following model was run by means of two-way ANOVA in r:*Body length of selected zooplankton species* = a + b_1_ • *Pelagic fish density* + b_2_• *Zooplankton species* + *e*

Parameters a and b are estimated by means of linear regression, and parametric statistics was used as zooplankton body length (of females) were assumed normally distributed.

## 3. Results

### 3.1. Fish Density

The estimated pelagic fish (larger than approximately 4 cm) density ranged from 34 to 4720 fish/ha (i.e., more than 100×), and estimated biomass ranged from 1.0 to 232 kg/ha (Table 4). The lowest density was recorded in the strongly regulated Lake Dokkfløy with whitefish as the pelagic species, and the lowest biomass was in the shallow Lake Furusjøen with Arctic charr as the pelagic species.. The highest density and biomass were recorded in the mesotrophic Lake Hemnessjøen with smelt and cyprinid dominance.

Echosounding was conducted repeatedly in five lakes, and among those, Lake Storsjøen in Rendalen and Lake Sølensjøen were surveyed in early (June/July) and late summer (August/September) within a year. The early summer estimates were substantially lower than those in late summer and demonstrated the importance of time of year for the recording. In Lake Storsjøen, there was a three-fold increase from June to August during the night, whereas the estimate at daytime in May 1986 showed an even higher figure than in August 1985, suggesting that a higher proportion of the stock was registered in May. In Lake Sølensjøen, it was estimated that there were 600 fish smaller than approximately 6 cm in June, probably mostly young of the year whitefish, whereas the density of fish larger than approximately 8 cm increased from 26 in June to 89 fish/ha in September. After these experiences, later echosoundings in coregonid dominated lakes were usually performed at daytime in spring, except for lakes at high altitude, which are not available at that time (i.e., Dokkfløy, Gopollen, and Sølensjøen), where nighttime in late summer was preferred.

Comparison across years in four lakes showed some interesting patterns. Estimates from 1985 and 2013 in Lake Engern suggested stability, whereas those from Lake Einavatnet, Lake Osensjøen, and Lake Storsjøen in Rendalen showed pronounced density increases during the last 10 to 20 years. In Lake Storsjøen, the increase coincided with illegal stocking of smelt and consequently reduced *L*_M_, whereas the increase in Lake Osensjøen was accompanied with reduced fish size, shown by the lowered *L*_M_ of fish species that have been present for more than a hundred years. In Lake Einavatnet, no special incident is known to have occurred during the density increase.

The pelagic *density* (fish of approximately 4 cm and larger), only including the “original” pelagic density from lakes that showed density increase (i.e., the highest estimate from the early years of lakes that were surveyed repeatedly, mentioned above), was positively correlated with *TP* (*r*^2^ = 0.72, *F*_15_ = 39.4, *p* < 0.0001) and with *conductivity* (*r*^2^ = 0.57, *F*_15_ = 19.9, *p* < 0.001), explaining 72 and 57%, respectively, of the fish density variation (Figure 2). The confidence intervals were admittedly large. The estimates of biomass kg/ha were also positively, although less significant correlated to *TP* and *conductivity* (*r*^2^ = 0.48, *F*_15_ = 13.8, *p* < 0.01 and *r*^2^ = 0.26, *F*_15_ = 5.4, *p* < 0.05, respectively). Two-way ANOVA with *TP* and median length (*L*_M > 4 cm_) as predictors gave a significant positive effect of *TP* (*F*_1,13_ = 89.2, *p* < 0.0001), non-significant effect of *L*_M > 4 cm_ (*p* > 0.05), and a significant (*F*_1,13_ = 19.8, *p* < 0.001) negative effect of the interaction *TP*:*L*_M > 4 cm_, revealing that high density corresponded to smaller fish in lakes with relatively high *TP*. There was no significant effect of altitude (*p* > 0.05).

The lakes were grouped in four categories according to the density estimates, where lakes with repeated recordings were grouped according to the “original” state: Category I included eight lakes with 34 to 186 fish/ha, Category 2 included five lakes with density from 269 to 610 fish/ha, Category 3 included three lakes with density from 1445 to 1841 fish/ha, and Category 4 included only the outlaying Lake Hemnessjøen with the highest density (4720 fish/ha) and biomass (232 kg/ha) estimates. The increased density over years in Lake Einavatnet moved it from Category 2 to Category 3, whereas the mean (1986 to 1998) estimate from Lake Osensjøen fitted in Category 1, and the 2011 and 2018 estimates fitted in Category 2. The increased estimates from 1985–2013 to 2016 in Lake Storsjøen in Rendalen moved it from Category 1 to somewhere between Category 2 and 3.

The length distributions, exemplified by 12 lakes (Figure 3a,b) showed that among the Category 1 lakes, a length group of about 20 cm dominated in Lake Atnsjøen with Arctic charr as the pelagic species, whereas there were small peaks in the range from approximately 5 to 13 cm, assumed to represent 0+, 1+, and 2+ fish. In the most elevated lake, Lake Rondvatnet, peaks were less pronounced, but were indicated about 5, 8, and 13 cm, otherwise, the frequencies decreased gradually to 40 cm. In the whitefish dominated lakes of Category 1, 79% of the fish were longer than 30 cm in Lake Engersjøen, whereas in Lake Sølensjøen, there was a higher proportion of smaller/younger fish, and 19% of the pelagic fish were longer than 25 cm. This difference was probably due to higher exploitation in Lake Sølensjøen. In Category 2 lakes, there were peaks in the length distributions between 8–10 and 20 cm, although Lake Randsfjorden showed a peak of fish larger than 30 cm, i.e., whitefish.

In the most species-rich lakes, the density was higher, and the group of small fish may include several cyprinid species. The size distribution of pelagic fish of Category 3 and 4 lakes (>1200 fish/ha) was dominated by fish smaller than 20 cm.

### 3.2. Zooplankton

The largest zooplankton species of the genus *Daphnia* was scarce or absent in lakes with high fish density (i.e., lakes dominated by smelt, vendace, and cyprinids). *Bosmina longispina* and *Daphnia galeata* were more abundant in the Category 1 and 2 lakes, whereas *D. cristata* were more important in the Category 3 and 4 lakes. Other species of *Bosmina* (*B. longirostris* and *lilljeborgii*) were important in Category 3 and 4 lakes. *D. longispina* and *D. lacustris* have only been recorded in Category 1 and 2 lakes. *Body length* of *Daphnia* and *Bosmina* were significant negatively correlated with *fish density* (Figure 4), and two-way ANOVA revealed significant effects of *fish density* (*F*_1,33_ = 92.7 *p* < 0.0001), and of *plankton species* (*F*_2_,_32_ = 171.3, *p* < 0.0001). According to the simple regression models (Figure 4), *D. galeata* fell short of 1.0 mm body length (= *very strong* fish predation, [63]) when pelagic density exceeded 1000 fish/ha, and *B. longispina* fell short of 0.48 mm (= *very strong* fish predation) at pelagic density above 2300 fish/ha.

## 4. Discussion

The pelagic fish density of the surveyed lakes increased with *TP*, and most of the fish were smaller than 15 cm of length in the most productive lakes, whereas in the less productive lakes, substantial numbers of fish were larger than 20 cm and even larger than 25 cm in lakes dominated by whitefish. In Arctic charr dominated lakes, there were many fish smaller than 15 cm, assumed to be immature fish.

The variation in estimated density within a year in Lake Storsjøen in Rendalen and Lake Sølensjøen demonstrated one weakness of echosounding as a method; the spatial distribution of the fish is important, and it varies through the season. The fish should dwell some meters below the surface and above the bottom, and not be distributed near the shores (bethic or littoral) to be registered. The highest fish densities were recorded in spring in some lakes, or in summer when zooplankton abundance was highest, in others. In Lake Sølensjøen, the whitefish stock was estimated by means of mark-recapture experiments in 1985–86 and 1993, and in 1986, the abundance of whitefish larger than 30 cm was estimated to N = 41 fish/ha [54], and in 1993, the density of whitefish larger than 33 cm was estimated to be 85 fish/ha [6]. This was roughly two to five times as high as the echosounding estimates of pelagic fish (which also included Arctic charr). Echosounding will always give a minimum estimate although it reflects the pelagic density and the potential for predation pressure on zooplankton, but this also depends on the timing.

In Lake Atnsjøen, there were also some fish larger than 40 cm, which were most probably piscivorous brown trout. Arctic charr and whitefish dominated lakes, like the Category 1 and 2 lakes, commonly harbor large brown trout feeding on small individuals of salmonids [69]. Smelt and vendace have limited prevalence compared with whitefish due to their immigration history [31], and the illegal stocking of smelt in Lake Storsjøen in Rendalen led to a rapid increase in pelagic fish as well as piscivore brown trout [30,70], demonstrating the smelt’s effect on piscivorous fish.

High fish density coincided with the scarcity or absence of large *Daphnia* species (i.e., *D. longispina* and *D. lacustris*), and with smaller body size of *D. cristata* and *D. galeata*. Body size of the widespread *Bosmina longispina* was also negatively correlated with pelagic fish density. This is in accordance with the findings of Hessen et al. [71], showing a negative relationship between the body size of zooplankton and the number of zooplankton feeding fish species present in the lake.

Sanni and Wærvågen [20] reported a fivefold increased abundance of *D. galeata* in the shallow Lake Mosvatn in southwest Norway after removing 100 kg of whitefish/ha by means of rotenone. The chlorophyll-a concentration in the experimental lake was reduced from 23 to 7 µg/L, Sechi depth increased from 1.7 to 2.3 m, and *TP* was reduced from 44 to 23 µg/L after two years, without reducing the phosphorous load. This experiment demonstrated a top down effect on the feeding chain and the lake’s trophic state with whitefish at the top of the chain. Several other experiments have led to similar results, although they have not always been successful [72,73]. Bottom-up control has also been demonstrated. Løvik and Kjellberg [74] found that *Daphnia* and *Bosmina* abundance in Lake Mjøsa decreased with decreasing *TP* [75] following comprehensive measures to reduce supplies from households, industry, and agriculture in the 1970s [76]. As primary production in freshwater is commonly limited by phosphorus [77], the scarcity of this element may indirectly limit the herbivorous zooplankton species if predation from fish is low or moderate. When phosphorus load is high, nitrogen may become scarce, favoring nitrogen fixing cyano bacteria, and some of these may produce toxins and represent the most serious threat to the public use of the water for bathing and fishing, etc.

Both Arctic charr [59], whitefish, vendace, and smelt [78,79] are shown to select large individuals of *Cladocera*. In Lake Mjøsa, vendace, whitefish, and smelt selected the larger *D. galeata* and the smaller *B. longispina* before *D. cristata*, which seemed to be avoided [78], whereas in Lake Ruskebukta in northern Norway, vendace and whitefish both fed on *D. cristata,* and the body size of the cladocerans was negatively related to fish abundance, especially of the plankton specialist vendace [79,80]. The predation was probably more intense in Lake Ruskebukta than in Lake Mjøsa due to the newly invaded vendace, whereas *D. galeata* was absent, and the smaller *B. longirostris*, compared to *B. longispina*, was important, possibly as an effect of predation. The body size of these species in Lake Ruskebukta was less than 0.81, 0.49, and 0.38 mm, respectively, and suggested *very heavy* predation pressure according to Løvik’s indices of predation pressure [45,63,81]. Løvik’s predation indices seem to fit the densities recorded in the present study, with the characteristic *market* predation pressure when *Daphnia* and *Bosmina* were shorter than 1.5 and 0.74 mm, respectively, and the characteristic *very heavy* predation when they were shorter than 1.0 and 0.5 mm, respectively. According to the regression equations of cladocera body size on fish density in this study, the characteristic *very heavy* predation occurs when the density of pelagic fish exceeds 1000 fish/ha for *D. galeata*, whereas the model for *B. longispina* predicts *market* to *strong* predation at this density (i.e., *Bosmina* are less vulnerable to fish predation). *D. cristata* are smaller than *D. galeata* and do not fit as well to the index as they are commonly smaller than 1.0 mm, though when considering the *Daphnia* genus, occurrence of the smaller *D. cristata* in lakes with heavy predation pressure fits well. The slope of the *D. cristata* model was similarly to that of the *B. longispina* model, suggesting similar vulnerability. High density and predation pressure occurred with the presence of smelt and vendace, which were mainly in Category 3 and 4 lakes, differing particularly from Category 1 lakes, dominated by Arctic charr and whitefish.

The density increase in Lake Storsjøen in Rendalen brought the lake from Category 1 to Category 2–3, and Lake Osensjøen ended in Category 2 due to increased density and reduced size and growth of vendace [8]. This must have led to increased predation pressure on zooplankton and probably caused a reduced size of the measured zooplankton species in these two lakes. Increasing abundance of perch and roach in Osensjøen [82] and Storsjøen in Rendalen [83,84], possibly due to increased temperature, may in the future add to the predation of zooplankton aside from the species dominating at present. Nevertheless, these two lakes are oligotrophic, and a reduced predation on algae from zooplankton can hardly lead to algal blooms. The proportion of cyanobacteria made up 0–0.2% of the total algae biomass in these lakes in 2011 [45].

Two lakes, Lake Einavatnet and Lake Storsjøen in Odalen, have been questioned as to whether they may develop to mesotrophic and cyanobacteria have been found to comprise 5–10% [40] and 20–25% [43,45], respectively, of the total algae biomass in events during summer. Both lakes harbor smelt and roach aside from whitefish, and the present situation may be due to improved recruitment and population growth, similarly to what took place in Lake Osensjøen between 2000 and 2010 [8].

In Lake Øymarksjøen and Lake Rødenessjøen of Category 3 and Lake Hemnessjøen of Category 4, several events of algae blooms have occurred in periods of sunshine and high temperature, especially after periods of flood and high (clay) particle load from the catchment. However, there is normally no oxygen deficiency in deep areas of the lakes [85]. Paleolimnological surveys suggest that algae blooms have occurred for hundreds of years and may be natural due to the soil characteristics of the catchment [86,87]. Unfortunately, there are no quantitative samples of zooplankton available from these lakes, but the species and body size of measured specimens suggested *very heavy* fish predation. Smelt and vendace are important in the pelagic zone of these lakes, and the presence of cyprinids [88,89,90] also adds to the predation on zooplankton as they are quite effective zooplankton feeders [91]. The ecology of these three lakes is slightly more complicated than in the other studied lakes because they harbor some larger species of crustaceans, *Gammarachantus loricatus, Mysis relicta*, *Pallasea quadrispinosa*, and *Ponteporeia affinis* [33], which can feed on smaller crustaceans.

## 5. Conclusions

Echosounding for the estimation of density and size distribution of pelagic fish in lakes is a useful method to monitor the pelagic zone in freshwater lakes. The spatial distribution of fish varies with season and time of day, and this should be tested in each lake before deciding the time of performance. The fish species composition of the lake also affects the distribution. The pelagic fish abundance increased by *TP*, not surprisingly, and high density was accompanied by small sized fish. The density showed pronounced increases during the last 10–20 years in three of the 17 studied lakes. The occurrence of herbivorous zooplankton (concentrating on cladocerans) was affected by fish density with regards to the species and size of species, suggesting predation effects. Lake monitoring by means of echosounding and exploring of species composition and body size of herbivorous zooplankton should be recommended, preferably in combination with quantitative algae and zooplankton sampling.

## Figures and Tables

**Figure 1 sensors-21-03391-f001:**
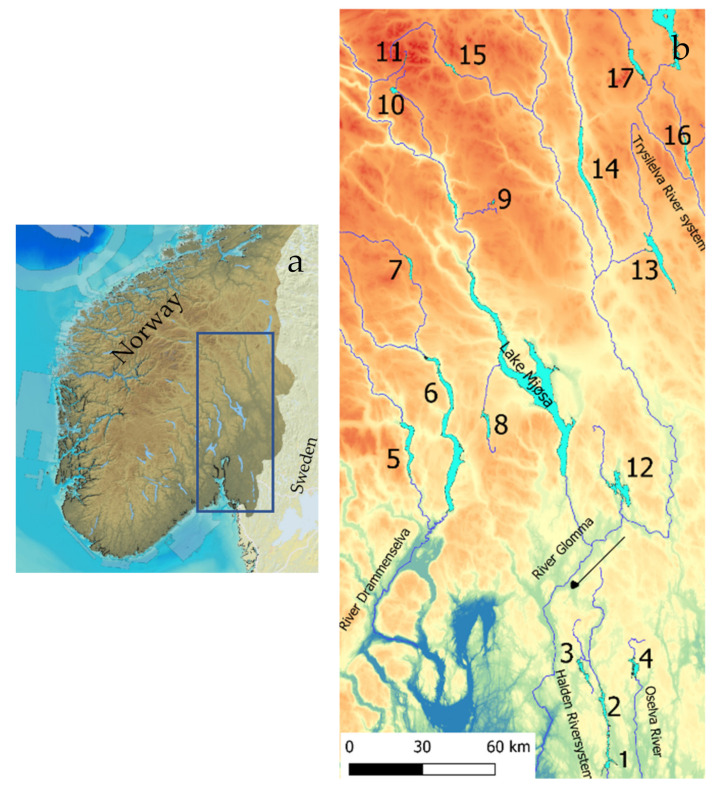
South Norway with the study area marked (**a**) and the river systems with numbers referring to Table 1, indicating the studied lakes (**b**).

**Figure 2 sensors-21-03391-f002:**
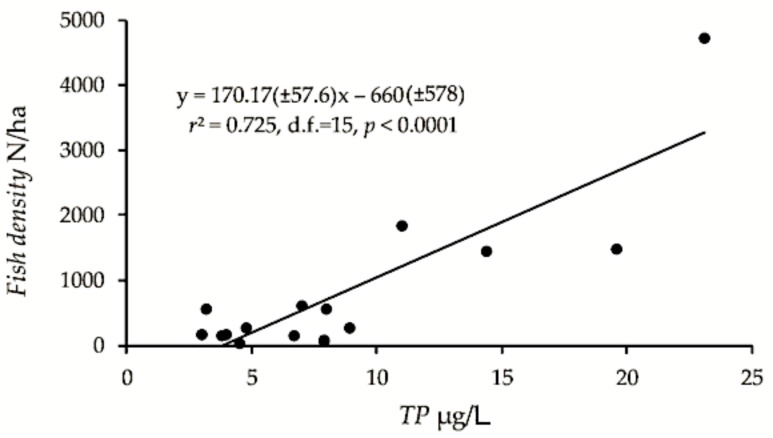
Pelagic fish density of the 17 studied lakes plotted on concentration of total phosphorus (*TP*), and the regression model with confidence intervals (±95% CI).

**Figure 3 sensors-21-03391-f003:**
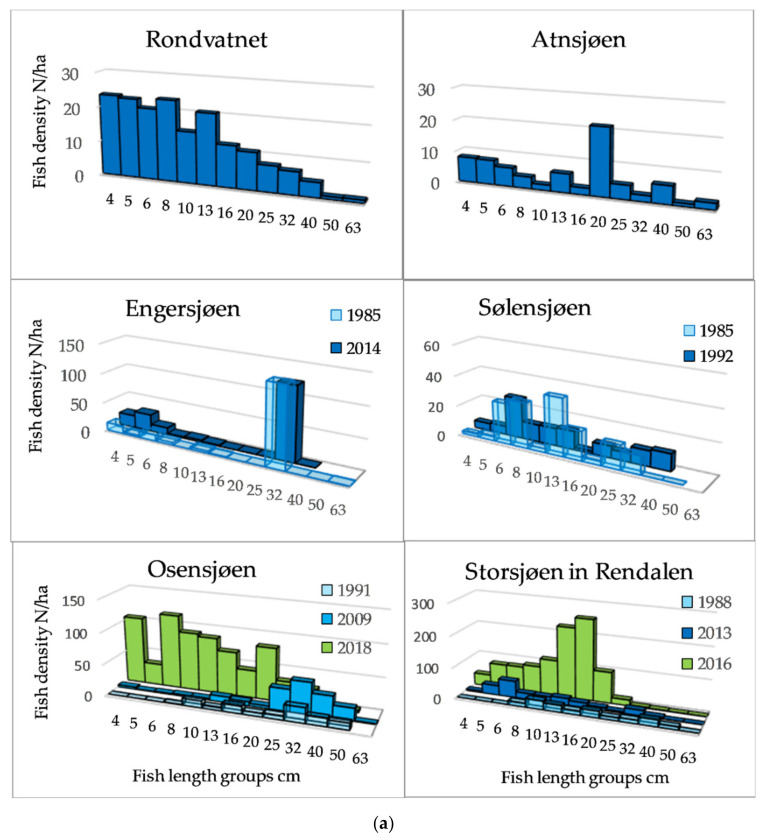
(**a**) Length distribution based on echo strength from pelagic fish in six of the studied lakes of Category 1 and 2. (**b**) Length distribution based on echo strength from pelagic fish in six of the studied lakes of Category 2, 3 and 4.

**Figure 4 sensors-21-03391-f004:**
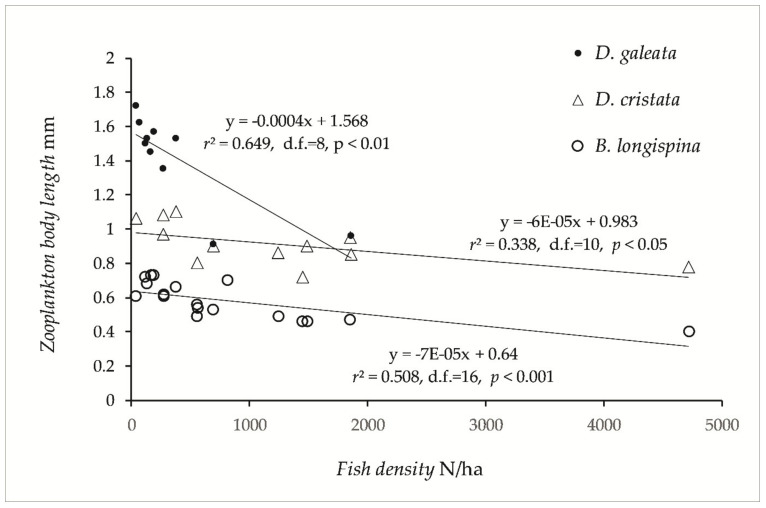
Mean length of three species of Cladocera plotted on pelagic fish density in the 17 studied lakes.

**Table 1 sensors-21-03391-t001:** Physical and chemical description of the 17 studied lakes. References for chemical data are given in Table 2.

	Lake	M o.h.	Surface	Regulated	Depth_Max_	Depth_Mean_	Sechi d.	*Cond.*	*TP*	*Tot N*
		m	Km^2^	m	m	m	m	mS/m	µg/L	µg/L
1	Øymarksjøen	107	13.6	1.0	35	16	1.6	5.4	14.4	867
2	Rødenessjøen	118	15.5	0.9	47	20	2.1	5.4	19.6	874
3	Hemnessjøen	133	12.7	1.4	35	10	1.6	7.1	23.1	494
4	Rømsjøen	138	13.7	0	100	24	4	3.3	8	415
5	Sperillen	150	37.3	2.3	129	43	5.0	2.2	4.8	314
6	Randsfjorden	135	140.7	3.2	131	52	7.0	3.9	8.8	389
7	Dokkfløy	735	9.5	65	53 ^1^	-	5.0	1.6	4.5	266
8	Einavatnet	398	13.7	2.3	56	-	5.5	9.0	7	1200
9	Gopollvatnet	982	1.47	2.2	20 ^1^	-	4	1.7	3.2	-
10	Furusjøen	852	5.3	0	23 ^1^	-	7	1.6	3.8	117
11	Rondvatnet	1167	0.96	0	55	-	6	0.42	3	149
12	Storsjøen Odal	130	44.3	0	17	7	3.5	2.6	11	390
13	Osensjøen	439	45.1	6.6	117	37	4	1.8	7.9	265
14	Storsjøen Rendal	251	48.1	3.6	309	145	5	4.1	6.7	390
15	Atnsjøen	701	4.8	0	80	35	7.5	0.78	7	177
16	Engeren	472	14.5	0	80	-	5	3.7	5.3	168
17	Sølensjøen	688	22.6	0	58	-	5	1.3	4	165

^1^ = deepest point recorded during fish assessment.

**Table 2 sensors-21-03391-t002:** Available data on algae- (wet weight Ww) and zooplankton (dry weight Dw) biomass in the studied lakes, and relative abundance of some important herbivorous zooplankton species/groups.

		Algae	Zooplankton	Approximately Biomass Distribution of Herbivorous Plankton Groups		
Lake	References	mg Ww/m^3^	g Dw/m^2^	20–50%	10–20%	<10%
Øymarksjøen	[32]	-	-	*B. coregoni, B. longirostris*	*B. coregoni, B. longispina*	
Rødenessjøen	[32,33,34]	281–1050	-	*B. coregoni, B. longirostris*		*B. longispina*
Hemnessjøen	[33,34]	1200–1818	-	*D. cristata*	*B. coregoni, B. lilljeborgii*	*B. lingispina*
Rømsjøen 1988	[35]	68–478	0.06–0.13	*Bosmina spp.*		
Sperillen	[36,37]	103–349	0.3–0.5	*D. galeata, B. longispina*	*H. gibberum*	*E. gracilis, D. cristata*
Randsfjorden 88	[38,39]	47–158	0.4–1.5	*E. gracilis*	*D. galeata, B. longispina*	*H. gibberum. D. cristata*
Dokkfløy 1998/2004	[38,39]	64–164	0.2–2.8	*D. galeata, H. gibberum*	*D. cristata, B. longispina*	*D. longispina*
Einavatnet 1988–2000	[35,40]	207–518	1.6–2.5	*D. galeata, B. longispina*	*D. longispina, H. gibberum*	
Einavatnet 2013	Unpubl.	-	-	*D. cristata*	*D. galeata*	*D. longispina*
Gopollen	Unpubl.	-	-	*D. cristata,*	*B. longispina*	*H. gibberum*
Furusjøen	[41]	-	-	*D. lacustris, B. longispina*	*H. gibberum*	
Rondvatnet	[42]	-	-	*B. longispina*		
Storsjøen Odalen 1988–2013	[43,44,45]	634–734	0.8		*B. longispina*	*D. cristata, H. gibberum, B. coregoni*
Osensjøen 1988	[35]	88	1.0		*H. gibberum*	*D. galeata*
Osensjøen 2005	[46]	-	0.7–1.7	*D. cristata, B. longispina*	*D. longiremis*	*D. galeata, H. gibberum*
Osensjøen 2011	[35,45]	133–238		*D. cristata, B. longispina*	*D. galeata*	*D. longiremis*
Storsjøen Rendalen 1985	[47,48]	38–247	0.6–0.9	*B. longispina*	*D. galeata*	*D. cristata, H. gibberum*
Storsjøen Rendalen 2011	[45]	466–1034	-	*B. longispina*	*D. galeata*	*H. gibberum*
Storsjøen Rendalen 2016/17	[44]	178–370	-	*B. longispina*	*D. galeata*	*D. longiremis*
Atnsjøen	[49,50,51]	143–180	1.7	*B. longispina*	*D. longispina, H. gibberum*	
Engern 1983	[52]	103–286	0.1–0.4		*B. longispina, D. galeata*	*D. cristata*
Engern 2007/2011	[45]	143–335	0.7	*D. galeata*	*B. longispina*	
Sølensjøen	[49,53]	116–247	-	*Eubosmina longispina*	*D. galeata, H. gibberum*	

**Table 3 sensors-21-03391-t003:** Body length of selected zooplankton species of the genera *Daphnia* and *Bosmina* species and pelagic fish species in the studied lakes.

Lake	References	*Daphnia* spp.	*D. Galeata*	*D. Cristata*	*B. Longispina*	*Bosmina* spp.	Pelagic Fish Species
Øymarksjøen	Unpubl.	-				0.46	Vendace, smelt, bleak, roach, bream, white bream roach, bleak
Rødenessjøen	Ref. [33]	-		0.90		0.46	Vendace, smelt, bleak, bream, white bream, roach
Hemnessjøen	Ref. [33]	-		0.78	0.48		Smelt, bleak, bream, white bream, roach
Rømsjøen	Unpubl.	-				0.50	Smelt, whitefish, vendace, roach, bleak
Sperillen	Unpul.	-					Whitefish, smelt, perch
Randsfjorden	Ref. [38]	-	1.35	1.08	0.61		Whitefish, smelt, perch
Dokkfløy	Ref. [38]	-	1.72	1.06	0.61		Whitefish
Einavatnet 2000	Ref. [40]	-		0.85	0.52		Whitefish, smelt, perch, roach
Einavatnet 2015	Unpubl.	-	0.96	0.87	0.49		Whitefish, smelt, perch, roach
Gopollen	Unpubl.	-		0.80	0.54		Whitefish
Furusjøen	Ref. [41]	1.81 ^1^			0.73		Arctic charr
Rondvatn	-	-	-	-	-		Arctic charr
Storsjøen Odalen	Ref. [43]	-		0.96	0.47		Whitefish, smelt, roach, bleak
Osensjøen 2005	Ref. [46]	-		1.13	0.63		Vendace, whitefish
Osensjøen 2011	Ref. [45]	-	1.53	1.10	0.66		Vendace, whitefish
Osensjøen 2018	Unpubl.	-	0.91	0.90	0.47		Vendace, whitefish
Storsjøen Rendalen 2011	Ref. [45]	-	1.50		0.72		(^4^ Smelt, sparse), whitefish, arctic charr,
Storsjøen Rendalen 2016	Ref. [44]	1.15 ^2^			0.66		^4^ Smelt, whitefish, arctic charr,
Storsjøen Rendalen 2017	Unpubl.	-	0.98		0.61		^4^ Smelt, whitefish, arctic charr,
Atnsjøen	Ref. [59]	1.62 ^3^			0.60		Arctic charr
Engern 2011	Ref. [45]	-	1.53		0.47		Whitefish, arctic charr
Sølensjøen	Ref. [45]	-	1.45	-	0.73		Whitefish, arctic charr

^1^ = *D. lacustris,*
^2^ = *D. longiremis*, ^3^ = *D. longispina*, ^4^ = introduced soon before 2010 [60].

**Table 4 sensors-21-03391-t004:** Survey lakes with time of survey (Year^Month Day/Night^), density (N/ha) with confidence interval (95% C.L), modal length including fish of approximately 4 cm length and larger (*L*_cm > 4 cm_), modal length excluding fish shorter than approximately 8 cm *L*_m > 8 cm_, and estimated pelagic biomass (B/ha, kg). Colors indicate lake categories.

Lake	Year	N/ha	95% CI	*L* _m > 4 cm_	*L* _m > 8 cm_	*B*/ha
Øymarksjøen	2014 ^10D^	1445	1116–1842	13.0	16	31.3
Rødenessjøen	2013 ^10D^	1486	1286–1670	13.0	13	41.8
Hemnessjøen	2018 ^10D^	4720	3956–5484	7.0	14	232
Rømsjøen	2014 ^10D^	553	433–638	12.6	13	17.4
Sperillen	2016 ^9D^	269	211–337	14.1	22	16.1
Randsfjorden	2015^9N^	270	162–970	8.0	12	21.3
Dokkfløy	2018 ^8N^	34	13–95	14.1	28	10.6
Einavatn	1990 ^5D^	610	494–726	7.9	13	41.8
	1996 ^5D^	1240	908–1573	10.0	13	44.7
	2013 ^5D^	1855	1193–2884	12.6	16	62.9
Gopollvatnet	2018 ^8N^	557	166–1617	12.6	20	54.5
Furusjøen	2017 ^8D^	153	58–396	5.0	11.5	1.0
Rondvatn	2017 ^8D^	169	42–662	8.0	16.0	9.5
Storsjøen Odal	2013 ^5D^	1841	1769–1914	10.0	13	28.4
Osensjøen mean	1986–1998^5D^	79	49–109	25.0	25	2.7
	2011 ^5D^	373	743–403	12.6	28.5	37.0
	2018 ^9N^	689	613–765	10.0	16	24.1
Storsjøen Rendalen	1985 ^6N^	26	22–29	5	25	3.6
	1985 ^8N^	79	62–96	5	25	6.0
	1986 ^5D^	116	88–145	25.1	32	28.2
	1988 ^5D^	151	131–171	16.0	20	27.3
	2013 ^5D^	109	92–126	8.0	16	9.4
	2016 ^5D^	809	749–870	13.0	16	21.2
Atnsjøen	2018 ^9N^	65	26–159	6.3	20	12.2
Engeren	1985 ^10D^	156	147–165	32.0	32	34.5
	2014 ^5D^	189	172–206	31.6	32	31.7
Sølensjøen	1985 ^7N^	107	94–120	10.0	20.0	7.1
	1992 ^6N^	464	341–587	4.0	20.0	3.8
	1992 ^9N^	159	124–194	13.0	16.0	6.6

## Data Availability

My methods were very non-invasive this time. When it comes to raw data, I intend to make them available, but I am not sure about how. I will anyway upload them on Researchgate with the paper.

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
