# Peer review of "Effects of Lake Productivity on Density and Size Structure of Pelagic Fish Estimated by Means of Echosounding in 17 Lakes in Southeast Norway"

_sensors, 2021, doi:10.3390/s21103391_

Round 1

Reviewer 1 Report

In this work an estimation of the pelagic fish density of several surveyed lakes is performed. They use echosounders to monitor the pelagic zone and verify the increased with TP in a rigorous statistical study including variation with season and time of day. I recommend this work for publication in Sensors after taking into consideration the following minor comments:

1-28. “The” is repeated

2-81. “The structure of zooplankton IN the community”?

8- 44. Show the degree of coverage S in the lakes that do not follow [61]

8-45. “Biomass was calculated”

8-46. Provide a reference or justification of the for the estimation W=a·L^b. Also for values of a and b for each species

8-56. Explain more detail the process: “the equipment was calibrated from the boat with the sphere”

8-63. An s is missing: “and it waS set to”

8.66. Explain how density was measured from the value of TS.

8.81.- y=ln(x)

13-181. Remove the repeated period in “was >25 cm.”

15-214. A p is repeated in “spatial”.

Author Response

Please see the attacted word document

Reviewer 2 Report

The study reported in this paper is laborious and important. Nevertheless, in some places, the description sounds more like a draft/report, without providing important details (especially related to the theoretical parts). For example, it is not clear how the specific parameters a and b are chosen (on page 8) and how sensitive is the estimated result to these values. In addition, the settings for PD and PRF could be better detailed. Similar issues related to the parameters a, b1, and b2 in the formulas that appear on page 9. In the reviewer opinion, Sections 2.2 and 2.3 would benefit from additional information regarding the issues mentioned before.

Another questionable issue is that the information about the “period of 34 years” appears in the Abstract, but it is never mentioned in the rest of the paper. This information is important in the overall context of the study and it should be better stressed out. On the other hand, in Section 3, there is a mentioned about 1985 (within the text, but also in Fig. 3a), which dates from 36 years ago. This aspect should be clarified.

Also, using mathematical symbols instead of words looks quite odd in some places. For example, see the excessive use of “>” and “<” instead of greater/longer or lower/shorter, respectively. For a draft or a report, it would be fine, but the abundance of these symbols (replacing words) within the sentences does not look good in a scientific paper.

Other issues:

- I would suggest rephrasing/splitting the last two sentences of the Abstract, which represent more than half of the Abstract. In the current form, they are quite long and difficult to follow.

- It is recommended to add one more paragraph at the end of Introduction, describing the paper organization (per each section).

- Please check the manuscript for typos. For example, see the beginning of the first line of Introduction, where there is an extra “the”, or the caption of Table 1, which is written as “Tabele 1”.

- There are also some grammar issues. For example, in the first sentence of Introduction:

and also serve as a human food --> and also serves as a human food

(The subject is “community”, which is singular, so the verb should be “serves”.)

- In the first sentence of Section 2.1:

The 17 study lakes --> The 17 studied lakes

- Similar issue in the captions of Figures 1-4 and Tables 1-3:

study lakes --> studied lakes

(alternatively, it could be “study’s lakes”)

- On page 6:

whitefish (Coregonus lavaretus) occur --> whitefish (Coregonus lavaretus) occurs

population size og whitefish were estimated --> population size of whitefish was estimated

- On page 8:

and it wa set to 0.32 ms --> and it was set to 0.32 ms

- On page 12:

and Category 4 include --> and Category 4 includes

- On page 13:

25 cm. . In Category --> 25 cm. In Category

- There is an extra comma in the middle of Fig. 4:

r^2 = 0.340,,

- On page 15:

the sppatial --> the spatial

?? in 198 and 1993

?? whitefish >.33 cm

- Page 22 is empty.

- The presence of the additional lines and numbers (from 2 to 15) in the header of pages 9-22 is not clear.

Round 2

Reviewer 2 Report

The author has addressed most of my concerns. However, it was not an easy task for the reviewer to assess the modifications performed in the revised version of the manuscript, mainly due to two reasons. First, the response of the author is very brief and without additional details or explanations. Most of the responses are in a short form, e.g., “CLARIFIED”, “ADDED DETAILS”, “TAKEN TO ACCOUNT”, etc. Second, the added text in the revised manuscript is not properly marked or highlighted. For example, see the case of the first comment, related to the parameters a and b on page 8. The response of the author is just “CLARIFIED”. But on page 8 we cannot find any marked/highlighted text related to the new explanations concerning the parameters a and b. The reviewer has to compare the two versions of the manuscript (initial submission and revised version) in order to figure out the differences.

Other issues:

- The mathematical symbols “>” and “<” are still used instead of the words “greater/larger” or “lower/smaller” in some sentences. For example, see on page 8, 2nd row: “was > 3” (this writing style could be used in a draft, but it should be avoided in a scientific paper). Similar issue on page 17, 8th row: “as it is commonly < 1.0 mm”.

- Also, on page 13, 3rd row: “by fish smaller than < 20 cm.” There is no need to use “<” in this case, since “smaller than” is already mentioned.

- A final proofreading of the manuscript is recommend, since there are still some language issues that should be revised.

Author Response

Dear Reviewer

Corrections were made on pages 8, 9, 13 and 17, and they are high lighted.

Best regards Arne Linløkken